# CRISPR/Cas9-Based Mutagenesis of Histone H3.1 in Spinal Dynorphinergic Neurons Attenuates Thermal Sensitivity in Mice

**DOI:** 10.3390/ijms23063178

**Published:** 2022-03-15

**Authors:** Zoltán Mészár, Éva Kókai, Rita Varga, László Ducza, Tamás Papp, Monika Béresová, Marianna Nagy, Péter Szücs, Angelika Varga

**Affiliations:** 1Department of Anatomy, Histology and Embryology, Faculty of Medicine, University of Debrecen, H-4032 Debrecen, Hungary; meszarz@anat.med.unideb.hu (Z.M.); kokai.eva@med.unideb.hu (É.K.); varga.rita@anat.med.unideb.hu (R.V.); ducza.laszlo@anat.med.unideb.hu (L.D.); szucs.peter@med.unideb.hu (P.S.); 2Department of Medical Imaging, Faculty of Medicine, University of Debrecen, H-4032 Debrecen, Hungary; papp.tamas@med.unideb.hu (T.P.); beres.monika@med.unideb.hu (M.B.); nagy.marianna@med.unideb.hu (M.N.)

**Keywords:** histone, dynorphinergic neuron, spinal cord, heat sensation, epigenetic regulation, pain

## Abstract

Burn injury is a trauma resulting in tissue degradation and severe pain, which is processed first by neuronal circuits in the spinal dorsal horn. We have recently shown that in mice, excitatory dynorphinergic (Pdyn) neurons play a pivotal role in the response to burn-injury-associated tissue damage via histone H3.1 phosphorylation-dependent signaling. As Pdyn neurons were mostly associated with mechanical allodynia, their involvement in thermonociception had to be further elucidated. Using a custom-made AAV9_mutH3.1 virus combined with the CRISPR/cas9 system, here we provide evidence that blocking histone H3.1 phosphorylation at position serine 10 (S10) in spinal Pdyn neurons significantly increases the thermal nociceptive threshold in mice. In contrast, neither mechanosensation nor acute chemonociception was affected by the transgenic manipulation of histone H3.1. These results suggest that blocking rapid epigenetic tagging of S10H3 in spinal Pdyn neurons alters acute thermosensation and thus explains the involvement of Pdyn cells in the immediate response to burn-injury-associated tissue damage.

## 1. Introduction

Neurons in the superficial dorsal horn (SDH) of the spinal cord are known to play a key role in the regulation of nociceptive information flow. Despite the increasing number of neuroepigenetic studies in pain research, the molecular mechanisms involved in the processing and differentiating of painful modalities are not well-characterized. While some post-translational modifications (PTM), e.g., histone H3.1 acetylation, in the maintenance of pathological pain have been already studied in detail [1,2,3], the role of other PTMs, such as phosphorylation at position serine 10 (S10) of histone H3.1 in nociception, has only recently been scrutinized (S10H3; p-S10H3) [4,5,6]. Based on our previous observations, phosphorylation of the histone H3.1 protein appears to be a reliable marker of enhancement of neuronal activity in the SDH following certain noxious stimuli [5,6].

General inhibition of S10H3 phosphorylation either by pharmacological blockage of the mitogen- and stress-activated kinases (MSK) [4] or by transgenic technology (using MSK1/2-gene-deficient mice) [5] prevents the development of heat hypersensitivity without affecting the development of mechanical allodynia following carrageenan-induced inflammation [5]. Several recent studies showed that somatosensory modalities are likely to be processed by definite but somewhat overlapping interneuronal populations in the SDH [7,8,9]. Of these inhibitory subgroups, dynorphinergic (Pdyn) neurons are regarded as primarily responsible for the gating of mechanical and pruritic pain [7,8,10,11]. We recently reported that spinal excitatory Pdyn neurons have a major contribution in the response to burn-injury-associated tissue injury via p-S10H3-dependent signaling [6]. Therefore, we hypothesized that specific blocking of histone H3.1 phosphorylation at position serine 10 (S10) in spinal Pdyn neurons alone would reduce or even eliminate central sensitization and heat hypersensitivity, consequently leading to a certain level of antinociception. In situ genetic manipulation of histone H3.1, however, is quite challenging due to its redundancy in the genome and its essential function during the mitosis of the cell cycle. To overcome this obstacle and test our hypothesis that p-S10H3 plays a critical role in the processing of noxious-heat-associated pain, in this study, we designed a recombinant-adenoassociated-virus (AAV9)-based expression vector that combines dominant-negative and CRISPR/cas9 technology. The vector induces expression of the mutant histone H3.1 in which serine is replaced with alanine at position 10 (S10A) and CRISPR elements for targeted deletion of the wild-type histone H3.1 genes. Intrathecal application of our AAV9 construct revealed that Pdyn neuron selective inhibition of S10 phosphorylation on histone H3.1 leads to a significant increase in the thermal nociceptive threshold, while leaving perception of other modalities intact.

## 2. Results

### 2.1. Distribution of Dynorphinergic Neurons in Various Brain Regions of the Pdyn::cas9-EGFP Hybrid Mouse

Somatic visualization of certain types of neuropeptides following standard immunostaining procedure often suffers from significant detection problems due to biosynthetic and trafficking characteristics of those molecules [12,13,14,15,16]. To overcome this problem in the case of dynorphin, several solutions have been developed and utilized recently. Antibodies raised against the dynorphin precursor preprodynorphin (PPD) in in situ hybridization (ISH) or even genetically engineered transgenic animals are all available to label cells selectively with a knocked-in fluorescent tag [6,15,17,18,19,20,21].

In the hybrid that we used throughout the study, Pdyn expression is linked to cas9-EGFP due to the cre-dependence of cas9. Thus, Pdyn could be identified based on their EGFP expression in this animal. To confirm the reliability of EGFP expression in Pdyn neurons, we used conventional immunohistochemistry on coronal sections of the whole brain (Figure 1 and Appendix A). Neurolucida reconstruction of the location of EGFP-positive cell bodies, after HRP-DAB conversion, proved that the great majority of DAB+ cells were restricted to those areas which have been designated as dynorphinergic-neuron-rich areas in the Allen Brain Atlas by ISH (Pdyn-RP_050505_04_B03-coronal. Available online: http://mouse.brain-map.org/gene/show/18376 (accessed on 25 August 2021); Figure 1) [22]. This finding confirmed that the hybrid selected for the experiments is indeed suitable for cas9-based manipulation of dynorphinergic neurons and for the selective insertion of S10A into Pdyn-expressing neurons exclusively.

In addition, we observed a high level of colocalization between the antibodies against the neuropeptide precursor preprodynorphin (PPD) and the EGFP signal in the superficial laminae of the spinal dorsal horn (SDH; Figure 2A) from a Pdyn::cas9-EGFP mouse, whereas in deeper laminae, where mainly excitatory Pdyn neurons are present [15,23], EGFP-immunoreactive neurons lacked Pdyn (Figure 2A).

Given that EGFP+/Pdyn-neurons were more numerous in the deeper dorsal horn laminae of the spinal cord in adult mice, those of neurons probably transiently expressed Pdyn at an earlier stage of their development. This might also be true for some of the supraspinal regions where abundant EGFP expression was detected. This hypothesis, however, could not be confirmed from the ABA (Figure 1 and Appendix A).

### 2.2. Validation of Our Experimental Strategy

Schematic representation of our experimental design and the final insert, which was synthesized and cloned into an adenoassociated viral vector serotype 9 (AAV9), are illustrated in Figure 2B,C, respectively. Prior to the osmotic pump implantation, the position of the intrathecal catheter was confirmed in the case of each animal with either microcomputed tomography (micro-CT) or conventional X-ray (see also in Methods; Figure 2D). Using RT-PCR, we confirmed the presence of the mCherry mRNA in the spinal cord, but not in the hippocampus of a wild-type mouse that had been transfected with the AAV9_mutH3.1 (Figure 2E), indicating that the intrathecal route of AAV9 delivery results in not only efficient but also spatially restricted transduction of spinal cord neurons (Appendix A). It must be noted, however, that RNA was isolated more than 3 weeks after post-transduction, and this may have led to a failure in detection of mCherry expression, driven by the relatively weak CMV promoter. Higher titer of the virus and RNA isolation at an earlier stage of infection, might have resulted in a detection of virus-coded mCherry signal at supraspinal areas.

To further confirm our strategy, we used anti-p-S10H3 antibody on lumbar spinal cord sections from Pdyn::cas9-EGFP mice transduced with AAV9_mutH3.1 (*n* = 3) and from non-transduced wild-type animals (*n* = 2). Prior to immunostaining against p-S10H3, mice were exposed to burn injury to induce S10H3 phosphorylation as detailed in our earlier report [6]. Image analyses revealed that there was a significant difference in the intensity profiles of p-S10H3-positive ROIs between the wild-type control (*n* = 98) and the AAV9_mutH3.1-treated hybrid mice (*n* = 145) suggesting that following CRISPR/cas9-based mutagenesis of histone H3.1 in spinal dynorphinergic neurons can indeed downregulate p-S10H3 (Appendix A). We also found an approximately 30% decrease in the amount of p-S10H3 immunopositive nuclei among the Pdyn-positive neurons in Pdyn::cas9-EGFP mice transduced with AAV9_mutH3.1 compared to the untreated Pdyn::cas9-EGFP mice [6].

### 2.3. Intrathecal Delivery of the Viral Constructs Effectively Transfects SDH Neurons

Viral infection in the SDH of Pdyn::cas9-EGFP mice was confirmed with immunocytochemistry, 5 weeks after the intrathecal administration. Transverse sections from the lumbar spinal cord of the sacrificed and perfused animals were immunostained with antibodies against GFP and RFP that recognized the genetically encoded EGFP and mCherry proteins, respectively (Figure 3).

Due to the cre-independency of mCherry expression in the AAV9_mutH3.1 vector, and the earlier reported tropism of AAV9 to glial cells when administered in the CSF [24], non-Pdyn neurons and even non-neuronal cells occasionally also showed mCherry expression in the sections.

Animals in the AAV9_mutH3.1-treated group showed a granulated-looking red fluorescence indicating mCherry scattered in the entire dorsal horn. However, the granulated staining was almost confluent in the cytoplasm of cells located in the more superficial region (laminae I–II) of the dorsal horn (Figure 3A). In deeper laminae (III–VI), the granulated mCherry signal was sparser resembling cytoplasmic dots around the nucleus labeled with DAPI (Figure 3A). This staining pattern is probably due to the lower expression rate of mCherry in this construct, which is a known issue when CMV promoter is used to drive protein expression in adult neurons [25]. 

Localization and pattern of the enhanced mCherry signal in animals of the AAV9_control-treated group exhibited an almost identical pattern to that observed in their littermates infected with AAV9_mutH3.1 (Figure 3B). The AAV9_control vector expresses mCherry as a fusion protein cre-dependently, thus, in theory, the mCherry signal should have been restricted to Pdyn-expressing neurons. However, mostly in the deeper laminae, some non-Pdyn cells still contained a weak mCherry fluorescence in a dotted fashion (Figure 3B) that is likely the result of leakage expression of this bicistronic fusion protein.

Sections taken from animals in the sham-operated group had no mCherry signal anywhere in the spinal cord (Figure 3C).

To determine the percentage of EGFP-positive Pdyn neurons that express mCherry in Pdyn::cas9-EGFP mice that had been infected with AAV9_mutH3.1 (*n* = 2 mice), papain-based single-cell suspensions from the lumbar segment of the spinal cord were analyzed by flow cytometry (FACS; Appendix A). In average, 1.45% ± 0.25 of the total number of cells/events exhibited EGFP-positivity (*n* = 2 mice). Of that, 52.6% ± 5.6 showed mCherry labeling suggesting that more than half of Pdyn neurons were infected by the AAV9_mutH3.1 based on their mCherry fluorescence (Appendix A).

### 2.4. Intrathecal Administration of AAV9_mutH3.1 Virus into Pdyn::cas9-EGFP Mice Increases the Thermal Nociceptive Threshold

Although prior works have reported that dynorphinergic neurons in SDH do not contribute to heat sensation in mice [7,8], our previous findings demonstrated that excitatory dynorphinergic neurons exhibited phosphorylation of S10H3 shortly after noxious heat-induced burn injury. Therefore, to evaluate their role in thermosensation, the thermal nociceptive threshold was determined using a hot plate with constant temperature (50 °C) in a set of in vivo experiments. AAV9_mutH3.1 or AAV9_control vectors were administered via intrathecal route into the subarachnoid space of Pdyn::cas9-EGFP hybrid and wild-type mice (Figure 4A,B). The thermal nociceptive threshold was measured before (BTM; baseline threshold measurement) and after the osmotic pump installation (days 7, 14, and 21; Figure 4A,B). 

The elevation in thermal nociceptive latency (represented as paw withdrawal latency; PWL) was most pronounced by the end of the first week (measured on day 7) after osmotic pump implantation in the group of AAV9_mutH3.1-treated mice (to 243.7% ± 28.4) as compared to the presurgery baseline (Figure 4A; Kruskal–Wallis ANOVA; details in Appendix A). The elevation, although less pronounced, remained significant throughout the 3-week observational period. At each time point after the infection, AAV9_mutH3.1-treated Pdyn::cas9-EGFP hybrid mice exhibited significantly higher thermal withdrawal latencies compared to values of the hybrid animals infected with the AAV9_control virus or the sham-operated groups (non-parametric Mann–Whitney test; see Appendix A). The average paw withdrawal latencies (PWL) in the AAV9_mutH3.1 treated mice before (BTM) and after (day 7) the infection was 15.0 ± 1.7 s and 33.8 ± 3.3 s (*n* = 7), respectively; while the corresponding values of the AAV9_control mice were 21.7 ± 1.1 s and 24.9 ± 1.3 s (*n* = 6), respectively. See also Appendix A for raw data.

Animals in the AAV9_control and sham-operated groups did not show significant alterations in PWL to noxious thermal stimulus compared to their baseline (BTM) values (Figure 4B; Appendix A; for statistical analysis see Appendix A). Like the other control groups, wild-type mice infected with AAV9_mutH3.1 only exhibited moderate alteration in PWL (*n* = 4). These data strongly suggest that S10H3 phosphorylation in Pdyn neurons is a crucial process for normal acute noxious heat sensation.

### 2.5. Intrathecal Administration of AAV9_mutH3.1 Virus into Pdyn::cas9-EGFP Mice Does Not Affect Mechanical Sensitivity

Several recent studies demonstrated that Pdyn neurons in the SDH of mice contribute to the suppression of mechanical sensation [7,8,10]. Thus, the mechanical withdrawal threshold (represented as PWL) was measured using von Frey filaments with increasing forces before and after the inhibition of S10H3 phosphorylation. PWL to noxious mechanical stimuli showed an increase on days 7 and 14 and reached 149.7% ± 12.0 by day 21, however, this alteration in tactile sensitivity was not statistically significant (Figure 4C), suggesting that blocking phosphorylation of S10H3 in Pdyn neurons did not affect mechanical sensation. Statistical analysis also revealed that there were no significant differences between the changes in AAV9_mutH3.1-treated hybrid mice compared to the corresponding values in the AAV9_control or sham-operated groups (see further details in Appendix A). There was a similar trend in PWL to mechanical forces applied to the paw in wild-type animals infected with the AAV9_mutH3.1 (*n* = 4). See also Appendix A in panel B for raw data.

### 2.6. Acute Chemosensation Was Influenced by the Viral Infection Itself but Mutant Histone H3.1 as Assessed by Formalin-Induced Nocifensive Behavior

Formalin test was chosen as a model to assess the effect of blocking phosphorylation of S10H3 on acute pain sensation [26,27]. Intraplantar injection of formalin (5%; i.pl.) evoked nocifensive behavior, in two phases depending on the participating peripheral components of the pain pathway [26,27]. In the first phase (0–15 min; direct activation of primary sensory neurons) the total duration of nocifensive reactions of animals in the sham-operated, AAV9_mutH3.1- and AAV9_control treated groups were 23.97 ± 2.0, 38.92 ± 2.6, and 38.53 ± 6.5 s, respectively (Appendix A). Interestingly, in the second phase of the test (15–60 min which corresponds to the effect of the inflammatory mediators released, the values in the sham-operated, AAV9_mutH3.1- and AAV9_control treated groups were 224.01 ± 32.7, 145.38 ± 2.6 and 146.73 ± 6.4 s, respectively (Figure 4D and Appendix A). Thus, while an almost 40% reduction was observed in the nocifensive responses among animals in the virus-treated groups (AAV9_mutH3.1. and AAV9_control) compared to the sham-operated animals, there was no significant difference between animals infected with the two types of AAV9 (Appendix A).

Thus, our data suggest that the response to formalin-induced acute pain was not affected by the S10A phenotype of histone H3.1 during the direct/early activation of sensory nerve terminals, nor during the release of acute inflammatory neuromodulators in the second phase.

### 2.7. Changes Related to the Mutant Phenotype of Histone H3.1 Were Not a Consequence of the Deterioration of the General Health of the Experimental Animals

To monitor the general well-being of mice after the surgery, body weight was followed for 3 weeks, performing measurements before each behavioral assessment. The body weight of animals decreased slightly at day 7 in both the AAV9_mutH3.1. (to 95.8% ± 0.7) and in the AAV9_control (to 97.1% ± 0.7) groups, with no significant difference between them (Figure 4E). By day 14 animals in both groups regained their baseline weight that was almost identical to those of the animals in the sham-operated group (Figure 4E; for statistical analysis see Appendix A). There was a similar tendency in post-operative weight change in the case of the wild-type animals that had been infected with the AAV9_mutH3.1 (*n* = 4). Interestingly, animals that received the AAV9_mutH3.1. virus intrathecally started to gain weight by the end of the third week, but the further investigation of this change was beyond the scope of this study and the experimental setup did not allow the follow-up of this weight change.

## 3. Discussion

We previously reported that Pdyn neurons had the largest share among p-S10H3-expressing neurons following burn injury both in wild-type and Pdyn::cas9-EGFP transgenic mice [6] suggesting their possible key involvement in the development of heat hyperalgesia after burn injury. Thus, in the present study, we tested the hypothesis that phosphorylation of S10H3 might be a crucial post-translational modification (PTM) in acute thermosensation by combining the AAV system, dominant-negative approach, transgenic technology, and CRISPR-based genome editing.

### 3.1. Validity of the Pdyn::cas9-EGFP Transgenic Model

Transgenic cre-recombinase-expressing driver mice strains occasionally show inhomogeneity in the expression level of the enzyme among CNS regions that, otherwise, are known to contain the type of neuron tagged with the cre enzyme. Therefore, we first tested whether our selected Pdyn-IRES-cre strain would be suitable for detecting Pdyn neurons in the spinal cord. The distribution of the reporter EGFP molecule in Pdyn::cas9-EGFP double-heterozygous progenies showed a close to identical expression pattern to the corresponding reference atlas (Allen Brain; Pdyn-RP_050505_04_B03-coronal. Available online: http://mouse.brain-map.org/gene/show/18376 (accessed on 25 August 2021)) in various regions of the brain [22]. Moreover, double-labeling experiments showed that the large majority of EGFP-expressing spinal dorsal horn cells in Pdyn::cas9-EGFP mice were also recognized by a specific antibody raised against Pdyn.

### 3.2. Cell-Specific Blocking of S10H3 Phosphorylation as a Precision Tool for Deciphering the Role of This PTM in the Complex Function of Pdyn Neurons

Histone H3 protein is encoded by twelve genes altogether in the mouse, which code for two isoforms: H3.1 and H3.2. Among those 12 genes, 4 produce H3.1 transcripts according to the HISTome2 database [28]. Post-translational modification, including acetylation, phosphorylation, and methylation of the core histone proteins, plays an important role in the epigenetic regulation of transcription in eukaryotes [29,30]. Phosphorylation of serine 10 in the N-terminal tail of histone H3 is not only involved in cell division in mitotic cells such as microglial cells but also participates in stimulus-dependent gene transcription in post-mitotic cells such as neurons [31,32], affecting only a certain subset of genes instead of the whole genome.

Given that this epigenetic tag is essential for survival [32], targeting histone phosphorylation with knockout technology is not an option. Due to using multiple copies of histone H3.1 genes [28] to generate a viable “partially” knockout mouse (i.e., point mutation at position serine 10 of histone H3.1), gene editing had to be designed in such a way that it would be restricted to only a distinct subpopulation of SDH neurons, in our case Pdyn neurons. The AAV-CRISPR system proved suitable for in vivo genome editing in a cell-specific manner in nondividing cells, such as postmitotic neurons [33]. Thus, neuron-specific mutagenesis of histone H3.1 at position S10 via viral-based in vivo genome editing was utilized to achieve relatively long-term expression of CRISPR components. Recombinant AAV vectors for transgene delivery offer several advantages, such as tissue-specific serotypes, lack of immune response, low toxicity, minimal integration capacity, and high transduction efficiency [33,34]. However, limited packaging size (up to 4.5 kb) of the viral vector and the transient nature of the transgene expression are regarded as disadvantages of the use of AAVs [33,34,35]. As we managed to solve packaging by using cas9-expressing transgenic lines, the major hurdle to the delivery of the multicomponent CRISPR complex was circumvented. With this approach, transgene expression is strictly controlled spatially on one hand by the cre-dependency of cas9 protein and on the other hand by administration of the AAV-CRISPR virus directly into the subarachnoid space through a limited interval via an osmotic minipump. We believe that these measures minimized the probability of off-target mutagenesis and undesirable side effects of this AAV9-CRISPR-based strategy in organs other than the spinal cord [33]. 

### 3.3. Technical Considerations

One important feature of our approach is the fact that cas9, whose expression otherwise is a rate-limiting step in the CRISPR/cas9 system, is available immediately after transduction with the virus due to the use of transgenic technology. It should be noted that long-term expression of cas9 may activate humoral immune response [35]. Although phenotypic signs of off-target effects were undetectable, target mismatch tolerance of CRISPR/cas9 cannot be excluded [36,37]. Together, these might contribute to the slight degree of weight loss at 7 days after administration of the viruses via osmotic pump implantation. Further screening of CRISPR off-target effects was beyond the scope of this study.

The CRISPR/cas9 system with homolog-directed repair (HDR) could not be used in our approach as HDR functions inefficiently in postmitotic neurons [38]. Therefore, we used a dominant-negative technology in combination with multiplex CRISPR/cas9-based genome editing for targeting the histone H3.1 gene in a single expression cassette. Using the multiple copies of the U6 promoter to drive sgRNAs, one cannot rule out genetic recombination in the viral sequence [39].

### 3.4. Putative Parallel Roles of Spinal Pdyn Neurons 

Blocking histone H3.1 phosphorylation in Pdyn neurons specifically by introducing the S10A mutation caused a significant and sustained elevation in the thermal nociceptive threshold that peaked at postoperative day 7. At the same time, neither mechanical sensitivity nor acute peripheral chemonociception to formalin was affected by blocking the same PTM. A robust confirmation of the role of Pdyn neurons was that wild-type C57/B6 mice injected with the AAV9_mutH3.1 showed no significant changes either in thermal or mechanical responses as S10A mutation only integrated into Pdyn neurons in a cre-dependent manner. 

Several recent reports concluded that spinal Pdyn neurons play a role in gating mechanical pain [7,8], while our results seemingly do not support these earlier findings. A possible explanation for this discrepancy comes from the different strategies by which the somatosensory dorsal horn neural network was targeted. With a chemogenic approach [7,8], the entire spinal inhibitory pool of Pdyn interneurons was ablated/erased from the spinal sensory circuits, which led to a reduced level of mechanosensation. In our experimental design, Pdyn neurons remained intact and fully functional in all other aspects except for a single PTM (phosphorylation of histone H3.1 on S10) and the consequential downstream target-derived actions. We recently found that the majority of p-S10H3-expressing dynorphinergic neurons were Lmx1b-IR (83.3%) in laminae I-IIo following burn injury [6], suggesting that noxious thermal pain is probably processed by a distinct excitatory subgroup of Pdyn neurons in the SDH. In line with our observation, a recent comprehensive transcriptomics study revealed that the glutamatergic subset of Pdyn neurons mediates hyperalgesia induced by peripheral tissue damage in rats [40]. Interestingly, it has been reported that thermosensitive neurons in the lateral parabrachial nucleus (LPb), a relay nucleus for ascending somatosensory pathway for pain, produce dynorphin [41]. Noxious cold, which fails to induce tissue injury, does not upregulate p-S10H3 expression in the spinal cord [5]. Therefore, the effect of this S10A histone mutation on noxious cold sensitivity was not tested. As the mutation induced by our strategy remained confined within the spinal cord, spinal Pdyn neurons can likely also participate in acute thermosensation through strictly controlled epigenetic regulation/machinery. Thus, it seems very likely that the reported Pdyn-dependent suppression of mechanical sensitivity [7,8] is regulated by other types of epigenetic mechanisms and that Pdyn neurons in the SDH contribute to the processing of more than one sensory modality while participating in the same anatomical circuit. 

### 3.5. Selective-Mutation-Based Fine Dissection of Complex Neuronal Functions—Future Perspectives for the SDH

Combining novel biotechnological tools, we provided evidence for the importance of histone H3.1 phosphorylation at position S10 in the Pdyn neuron in the response to painful thermal stimulation. In certain pathological pain states (inflammation or nerve injury), this neuroepigenetic signal may be one of the molecular mechanisms that results in increased neuronal activity and consequential hyperalgesia through permissive transcription of certain pain-related genes such as prodynorphin itself [21,40,42,43,44,45].

Pdyn neurons are members of the endogenous-opioid-releasing neurons that participate in the descending pain modulatory networks [42,46]. Thus, it is reasonable to suggest that similar epigenetic tags on the same or other motifs may be equally important in the regulation of parallel functions of other neuronal types within the same system, such as enkephalinergic neurons. 

A thorough description of the modulatory effects of different PTMs within the neurochemically discrete groups of SDH neurons upon different somatosensory modalities is a necessary but tedious task. Nevertheless, we believe that this approach would explain some of the contradictory findings concerning the function of certain neuronal populations in the SDH and might shed light on how complex sensory processing of different modalities is solved with a limited number and variety of neurons within the SDH.

## 4. Methods

### 4.1. Animals

Animal experiments were approved by the Animal Care and Protection Committee at the University of Debrecen (No.: 23-1/2017/DEMÁB) and were performed in line with the European Community Council Directives and the IASP Guidelines. Pdyn-IRES-Cre mice (see key resources provided in Appendix A) that expressed Cre recombinase under the direction of the Pdyn (prodynorphin) promoter [19] were crossed with Rosa26-LSL-Cas9 (Appendix A) knock-in mice with Cre-dependent expression of CRISPR-associated protein 9 endonuclease (cas9) and enhanced green fluorescent protein (EGFP) directed by a CAG promoter [20]. Genotyping of litters from both strains was routinely performed by PCR (for primer sequences, see Appendix A). In the resulting hybrid mice (Pdyn::cas9-EGFP), all Pdyn-containing neurons showed strong somatic EGFP expression in the brain (Figure 1) as well as in the spinal cord (Figure 2A; Figure 3). Adhering to the 3R principle (replacement, reduction, refinement), altogether 30 adult male mice were used (between 24.9 g and 34.1 g; 22 Pdyn::cas9-EGFP hybrids and 8 wild-type C57BL/6 mice). All the mice underwent surgical intervention, either cannulation for the osmotic pump or sham operation.

### 4.2. Designing the Construct Containing the Mutant Histone H3.1 and CRISPR Elements

The dominant-negative mutant S10A H3.1 coding sequence, multiplex single-guide RNAs (sgRNAs), and even a fluorescent reporter gene were incorporated into the all-in-one AAV construct. The dominant-negative sequence, including the complete histone H3.1 sequence with serine-to-alanine point-mutation at position S10 (S10A; mutH3.1) driven by a strong synthetic hybrid CMV enhancer/chicken β-actin (CBh) promoter, was incorporated into a pCBh cloning vector in silico using SnapGene software (Ver. No.: 5.1.0; see Appendix A). from Insightful Science; Available online: http://www.snapgene.com, accessed on 15 March 2022). Both ends of this template sequence contained loxP sites, ensuring cre-dependent transcription of mutH3.1. Thus, in addition to cas9 expression provided by the transgenic line applied, the expression of the mutant H3.1 template was also restricted to Pdyn neurons due to their cre dependency.

For multiplex CRISPR/cas9-based genome editing, a human U6 small nuclear promoter was chosen to express three different sgRNAs targeting the same gene (histone H3.1 gene) in a single-expression cassette. sgRNA sequences targeting wild-type histone H3.1 are shown in Appendix A.

Additionally, mCherry fluorescent coding sequence under the control of constitutive CMV promoter was also placed behind the mutH3.1_sgRNA cassette without adding loxP sites. This way, the cre-independent expression of mCherry allowed verification of the viral infection by detecting the immunohistochemically enhanced red signal with fluorescent microscopy. The final construct (Figure 2C), which contained the mCherry sequence and the mutH3.1_sgRNA cassette, was flanked by BamHI restriction sites at both ends. Its total length was 4399 bp. This insert was then synthesized and cloned into a commonly used cloning vector pUC57 by GenScript. Complete insert sequences with color codes can be found in Supplementary Data S1. MutH3.1-containing final insert in a pUC57 plasmid was packaged into serotype 9 recombinant adenoassociated virus vector (AAV9) in a pilot scale (15 × 150 mm plate) and purified with iodixanol for in vivo application by SignaGen Laboratories (Frederick, MD, USA; titer > 1 × 10^13^ VG/mL), resulting in AAV9_mutH3.1. After ultracentrifugation, viral titer (in VG/mL) was determined by titration via qPCR. 

### 4.3. Intrathecal Administration of the Viral Vector

Before the implantation pumps were soaked in sterile 0.9% saline for a couple of hours to promote the priming procedure, pumps were filled with the viral solution in a final volume of 100 µL in a titer of 3 × 10^9^ VG/mL under the fume hood using a cut-ended pipette microtip. Till the insertion of the osmotic pump, the pump reservoir was placed in an upright position into an Eppendorf tube to avoid evaporation of the solution from the pump. The release rate for the applied ALZET pump model (Cat. No.: 1003D; DURECT Corporation, Cupertino, CA, USA) was 1 µL/h, and the duration of the complete release was 3 days.

After induction of anesthesia using intraperitoneal administration of sodium pentobarbital (˂50 mg/kg), each mouse was placed in a prone position. The skin on the back of the animal was shaved and disinfected with 70% ethanol, and an incision in the midline was made. For intrathecal cannulation, the tip of a 26G needle was gently inserted at about a 90° angle into the L4-L5 intervertebral space at the midline. As soon as the tip of the needle passed through the dura mater, the characteristic tail-flick reflex could be observed. Next, the angle of the needle was decreased to about 30° and slightly pushed rostrally for 1–2 mm in the subarachnoid space. The needle was replaced by a polyurethane mouse intrathecal catheter including a Teflon-coated stylet for easier placement. After validating the correct position of the catheter in the intrathecal space using microCT or X-ray, the cannula was attached to the Alzet osmotic pump, which had been filled with the viral solution. The wound was sutured, and the mice were allowed to fully recover. Three days after the surgery, both the pump and the catheter were removed from the reanesthetized animals.

### 4.4. Control Groups

Three sets of controls were included in this study. In the first control group, referred to as “AAV9_control” throughout the text, Pdyn::cas9-EGFP mice (*n* = 6) were injected with ready-to-use AAV9 particles containing the humanized channelrhodopsin H134R mutant which was fused to mCherry, driven by the EF1a promoter (Appendix A). According to the manufacturer, the titer was ≥1 × 10^13^ VG/mL. The animals underwent the same surgical procedures (see above) and received the control virus in the same titer as the AAV9_mutH3.1 (3 × 10^9^ VG/mL). In the second control group, wild-type C57/B6 mice were treated with the AAV9_mutH3.1 (*n* = 4), while the third control group comprised sham-operated Pdyn::cas9-EGFP mice (*n* = 4). Animals in the sham-operated control group underwent the same surgical intervention as described above. The only difference was that the osmotic minipump had not been attached to the catheter. Anaesthesia, skin incision, insertion of the catheter, and even the reoperation on the third day to simulate removal of the catheter were the same.

### 4.5. Verification of the Position of the Intrathecal Catheter by 3D Microcomputed Tomography (Micro-CT)

The SkyScan 1272 compact desktop micro-CT system was used to determine the location of the intrathecal catheter in deeply anesthetized mice (Figure 2D), using the following scanning parameters: image pixel size, 26 μm; matrix size, 672 × 1008 (rows × columns); source voltage = 60 kV; source current = 166 μA; rotation step (deg) = 0.300, filter = Al 0.25 mm. Flat-field correction and geometrical correction were applied to the images. Scan duration: 0 h:28 min:19 s. Reconstruction of the cross-sectional images from tomography projection images was performed with the SkyScan NRecon software (version 2.0.4.2). Postalignment, beam-hardening correction, ring artifact correction, and smoothing were completed during postprocessing of the image data. The output formats were DICOM and BPM images. The 3D Volume rendering tool provided by RadiAnt DICOM Viewer (Medixant, Poznań, Poland) was utilized to visualize 3D micro-CT images (Figure 2D). Micro-CT validation of catheter position was performed in the case of the first 5 interventions. Since ionizing radiation was hinted to influence the experimental outcome [47], in the case of the remaining mice, the verification of catheter position was performed with conventional X-ray imaging to reduce the dose load. The average estimated dose load was 125 mGy (lethal dose in mice ranges are from 5.0 to 7.6 Gy, depending on the strain and age [48,49]. 

### 4.6. Thermal Sensitivity Assessments

Response latency to noxious heat (50 °C) was evaluated using a hot-plate test. Before testing, mice were preconditioned for the hot plate every day for a week before and on every second day after the surgical implantation of the osmotic pump. During preconditioning, animals were placed onto the hot plate, which was set to an innoxious surface temperature (37 °C), for approximately 10 min. Thermal response latency was determined by an independent observer who was blind to the treatment. When the animal exhibited discomfort upon constant high temperature (i.e., sudden lifting/withdrawal, licking, or shaking the affected hindlimb), the heating of the surface was immediately terminated, and the response latency (s) was noted. The maximum cutoff latency was set to 50 s to prevent burn injury of the paw. Response latencies to noxious heat were assessed in a three-day window before (BTM; baseline threshold measurement; Figure 4A) and on days 7, 14, and 21 after the osmotic pump implantation (Figure 4A,B). Since a distinct disadvantage of the hot-plate test is its sensitivity to repeated measurements, likely via learning [50,51,52], we only took a single measurement in the case of each animal at a given time point. The averaged response latencies of the animals on each measurement day were normalized to baseline latencies (the value shown for day 0) and displayed as percentages.

### 4.7. Mechanical Sensitivity Assessments

Tactile sensitivity was measured as paw withdrawal latency (PWL) to dynamic von Frey stimulation [53] at the same timepoints as the hot-plate tests. The maximum force used was 5 g (increasing between 0.8 and 5 g) with 10 s intervals between trials, performed by two independent observers who were blind to the treatment. The withdrawal latencies on each measurement day (day 7, 14, and 21 after the osmotic pump implantation; Figure 4A,C) were normalized to baseline withdrawal latencies (day 0) and displayed as percentages.

### 4.8. Formalin-Induced Acute Somatic Nocifensive Behavior

Following the 3-week postimplantation period, rapid inflammation-induced pain was evoked by injecting formalin (5% of Formaldehydum solution 37%; Ph.Hg. VII.; in a volume of 25 μL; i.pl.) into the right hind paw of sham-operated or Pdyn::cas9-EGFP hybrid mice which had been injected with one of the viruses (AAV9_mutH3.1or AAV9_control). Animals were then immediately placed into a 17 × 17 cm plexiglass observation chamber equipped with a mirror in the back, and formalin-induced nocifensive behavior was recorded with the camera, continuously for one hour. 

Nocifensive behavior was assessed as the length of time spent lifting, licking, or shaking the treated hind paw in 5 min periods across 60 min. Evaluation of the experiments was carried out with an experimenter blinded to the treatment. The contralateral leg was omitted from the observation. Formalin-induced somatic chemonocifensive response appears in two phases [26,27]. Due to transient and rapid activation of sensory nerve endings, the early phase lasts only 5–15 min, while the late phase (15–60 min) is generated by the release of acute inflammatory mediators [27]. The duration of nocifensive behavior was measured in both periods [26].

### 4.9. Tissue Preparation for Microscopic Analysis

Sampling and tissue processing were performed as described previously [6]. After inducing nonrecoverable anesthesia with sodium pentobarbital (50 mg/kg intraperitoneal), animals were transcardially perfused with 4% paraformaldehyde (PFA). The brain and the lumbar spinal cord were removed and sectioned with a vibratome (Leica CLS 100X; Wetzlar, Germany) at 150 and 100 µm thickness, respectively. 

### 4.10. Immunoperoxidase Staining

For mapping the supraspinal distribution of Pdyn neurons in Pdyn::cas9-EGFP mice, the whole brain was removed and sectioned. After quenching endogenous peroxidase activity with Dent’s bleach (methanol:DMSO:H_2_O_2_ in a ratio of 8:1:1), an antibody raised against GFP (Appendix A) was added to the samples (1:4000; 2 days). Two-hour-long incubation with the secondary antibody (1:500; room temperature) was followed by adding extravidin (1:500) to the specimen. All reactions were carried out overnight at 4 °C unless otherwise stated. Finally, the DAB peroxidase substrate kit was used for the visualization of the GFP-positive signal. Sections were counterstained with Toluidine blue and mounted in Eukitt mounting medium. Reagents are summarized in Appendix A. The location of the Pdyn neuronal somata containing DAB precipitate was noted with the Neurolucida software (v11.07) in randomly selected slices (*n* = 8), using 10× objective lens (Olympus, Tokyo, Japan).

### 4.11. Visualization of mCherry Expression in the Lumbar Spinal Cord by Immunofluorescent Staining for Confocal Imaging

Cells infected with the AAV9_mutH3.1 were identified based on their mCherry expression. Confocal microscopic analyses were performed on transverse lumbar spinal cord sections as detailed in our previous study [6] with slight modifications. Briefly, a primary antibody mixture was applied on sections (overnight; 4 °C), which contained chicken anti-GFP (1:2000) and rat anti-RFP (1:1000; see details in Appendix A) to enhance the native fluorescent signals. Cell-nuclei-specific DAPI and species-specific secondary antibodies raised in donkey conjugated to Alexa Fluor-488 or 555 (see in Appendix A) were added to the sections for 2 h at room temperature at the end of the immunofluorescence staining protocol. All antibodies were diluted in phosphate-buffered saline (PBS) supplemented with a 0.3% Triton-X 100. Sections were mounted in a Hydromount medium, and confocal images were scanned with Olympus FV3000 confocal systems.

Confocal images were acquired with the same settings (PMT voltage, laser transmissivity, Z dimension parameters, etc.) using 10× lens (UPlanSApo, Olympus, N.A. 0.4). Confocal image stacks consisted of 4 optical images at 3.6 µm z-separation. In some cases, for higher magnification with 40× objective lens (UPlanFLN, Olympus, N.A. 1.3), 16 optical sections of 0.5 µm thickness were acquired unless otherwise indicated. Postprocessing of the images was carried out with the FV31S-DT software (Ver. No.: 2.3.1.163).

### 4.12. Detection of mCherry mRNA and the Mutant Variant of Histone H3.1 Transcripts in the Spinal Cord

Five weeks after IT injection of rAAV9_mutH3.1 into a wild-type mouse, total RNA was extracted from the harvested lumbar segment of the spinal cord and from the hippocampus using TRIzol reagent (see in Appendix A). Total RNA was reverse-transcribed, and specific fragments (mCherry and GAPDH) from cDNA were amplified with DreamTaq DNA Polymerase (Thermo Fisher Scientific; Waltham, MA, USA) according to manufacturers’ recommendations. The primer sets were designed by using Primer3Plus and are shown in Appendix A. 

### 4.13. Dissociation of Spinal Cord Tissue to Single Cells for Fluorescence-Activated Cell Sorting (FACS)

Pdyn::cas9-EGFP and C57Bl6 mice were euthanized via Na-pentobarbital. The lumbar spinal cord was extracted, finely minced with a scalpel, and placed into Neurobasal medium supplemented with Glutamax and Na-bicarbonate (see Appendix A). Samples from C57Bl6 mice served as controls for FACS gating. Chopped tissues were incubated in the presence of 100 U papain and 0.25 mg hyaluronidase type I (see Appendix A) for 2.5 h at 230 rpm at room temperature (RT) in a total volume of 2 mL of Neurobasal medium. Trypsin-EDTA was added in a final concentration of 0.125% to the homogenates for 30 min. To stop digestion, 20% FBS was added to the supernatant of the cell suspensions. The resulting single-cell suspension was centrifuged at 300× *g* for 5 min at RT. Cell pellets were dissolved in cold phosphate-buffered saline (PBS; pH = 7.4) and placed on ice. Single-cell suspensions were then further homogenized by filtration through a 41 μm mesh filter. 

### 4.14. Flow Cytometric (FACS) Analysis

Papain-based single-cell isolation from the lumbar spinal cord of Pdyn::cas9-EGFP mice was followed by flow cytometry (FACS Aria III) using FACSDiva 6.1.3 software to determine the percentage of double-labeled (EGFP+ and mCherry+) Pdyn neurons. Osmotic pump filled with AAV9_mutH3 was implanted into Pdyn::cas9-EGFP mice for 76 h prior to FACS analyses (*n* = 2). Parameters were set to sort precision mode “Purity” to ensure a stringent sorting of double-positive neurons (EGFP+ and mCherry+). 

### 4.15. Evaluation of p-S10H3 Immunoreactivity after Burn Injury

In order to quantify to what extent the level of histone H3.1 phosphorylation decreases upon transduction with AAV9_mutH3.1, Pdyn::cas9-EGFP mice (*n* = 3) were injected intrathecally with AAV9_mutH3.1, and five days later burn injury protocol was applied as detailed in our earlier report [6]. Burn injury was followed by transcardial perfusion with 4% PFA, and then spinal cord sections from the exposed animals (*n* = 3) were double-immunolabeled against GFP and p-S10H3 [6] to determine the influence of AAV9_mutH3.1 on the level of burn-injury-induced S10H3 phosphorylation. Wild-type control mice were also subjected to burn injury, and their samples served as controls (*n* = 2). Images for analysis of immunopositive signals were captured by Olympus FV3000 confocal system using a 40× oil immersion objective (UPLFLN40XO, NA 1.3) to adjust the detectors for obtaining nonsaturated images (12 bits/px for each channel). The acquisition parameters were the same for capturing images. From each image, 20 consecutive confocal stacks (lateral resolution 0.31 µm/px, axial resolution 0.37 µm/px) were transformed to 2D images by using maximum intensity projection of the Olympus FV31S-SW software (Ver. No.: 2.3.1.163). The p-S10H3-immunopositive nuclei were then marked manually (ROI), and the fluorescence intensity profiles were measured with Fiji [54], and the average gray level values were compared between the transduced (AAV9_mutH3.1) and nontransduced animals.

### 4.16. Statistical Analysis

Detailed statistical analyses for the data presented in Figure 4 can be found in Appendix A. The PWL values and values for body weight were normalized to the baseline level (BTM; baseline threshold measurements; considered to be the same on day 0) and expressed as percentages. All normalized data were averaged and are presented in the relevant figures as mean ± standard error of mean (SEM) of *n* = 3–7 mice per group (see Appendix A). 

The Kruskal–Wallis nonparametric ANOVA test with Origin Pro9 software (64-bit, Ver. No.: 9.0.0) was applied to determine the overall significance of the treatment on paw withdraw latency in response to noxious heat/mechanical force or on body weight (raw values were applied for analysis).

With the aid of Origin Pro9, the Mann–Whitney U test was used to compare the three experimental groups (AAV9_mutH3.1; AAV9_control; sham group) at each measurement point (normalized values were applied for comparison). Additionally, the Mann–Whitney U test was used for comparing the average gray level intensity values of p-S10H3-immunoreactive nuclei on confocal images from transduced (AAV9_mutH3.1) and nontransduced animals (see Appendix A).

In the case of the formalin test, the overall influence of each treatment on the 1st and 2nd phases of formalin response was evaluated by the Kruskal–Wallis nonparametric ANOVA test.

In all cases, *p* < 0.05 or *p* < 0.01 were accepted as statistically significant as indicated in the relevant figures.

## Figures and Tables

**Figure 1 ijms-23-03178-f001:**
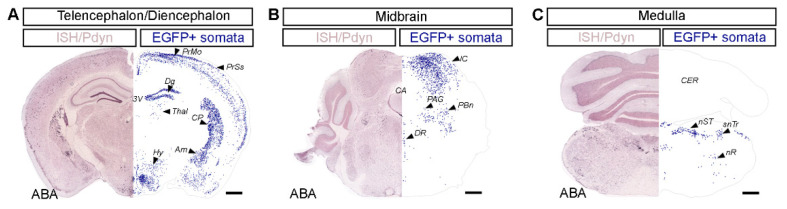
Immunohistochemical mapping revealed that regional distribution of dynorphinergic neurons in the brain of a Pdyn::cas9-EGFP mouse shows large similarity to the reference atlas provided by the Allen Institute. The distribution of Pdyn neurons was evaluated by plotting immunopositive cell bodies (revealed by the HRP/DAB method) through the telencephalon/diencephalon (**A**), the midbrain (**B**), and the medulla (**C**) of a Pdyn::cas9-EGFP mouse. On the right side of each panel, blue dots represent the location of Pdyn-immunopositive neurons, detected and reconstructed with the aid of Neurolucida (EGFP + somata). The left side of each panel shows the corresponding reference image of in situ hybridization (ISH) data from the Allen Brain Atlas (ABA) [22] displaying Pdyn mRNA expression pattern. PrMo, primary motor area; PrSs, primary somatosensory area layer 2/3 and layer 5; Dg, dentate gyrus; Thal, mediodorsal nucleus of the thalamus; Hy, dorsomedial nucleus of the hypothalamus; CP, caudoputamen; Am, central amygdalar nucleus; 3V, third ventricle; IC, inferior colliculus; PAG, periaqueductal gray; PBn, parabrachial nucleus; DR, dorsal nucleus raphe; CA, cerebral aqueduct; CER, cerebellum; nST, nucleus of the solitary tract; snTr, spinal nucleus of the trigeminal; nR, reticular nucleus. Scale bar, 500 µm. For original images, see Appendix A.

**Figure 2 ijms-23-03178-f002:**
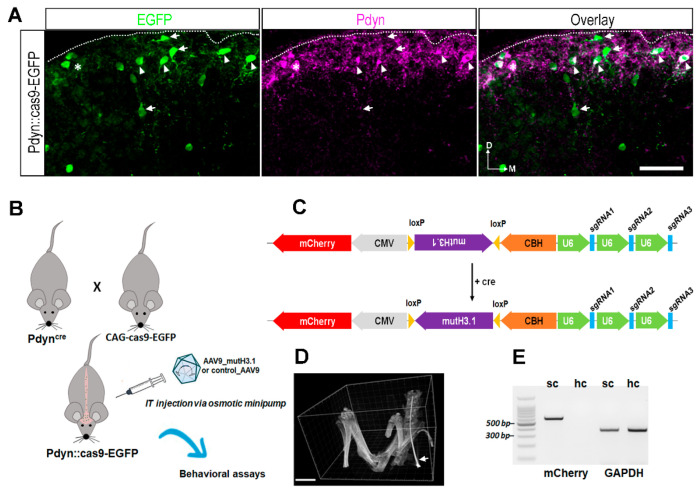
Cell-type specific targeted delivery of the necessary components for genome editing of histone H3.1 via CRISPR/cas9 strategy. (**A**) Immunostaining with antibodies against EGRP (green), Pdyn (magenta) in a projected image of seven optical sections with a 40× lens from a transverse spinal cord section of a Pdyn::cas9-EGFP mouse. The overlay shows a merged image. The numerous EGRP-immunoreactive neurons are predominantly visible in the superficial layers of the lumbar spinal cord. The majority of these are Pdyn+ (arrowheads), while some of them lack Pdyn (arrow). Few Pdyn-expressing neurons lack the EGFP signal (asterisk). D, dorsal; M, medial; scale bar, 50 μm. (**B**) Schematic drawing representing the CRISPR/cas9 strategy to establish the mutant histone H3.1 (mutH3.1) in Pdyn neurons. The abbreviation mutH3.1 refers to serine-to-alanine exchange (S10A) at position serine 10 of the wild-type histone H3.1. IT, intrathecal; AAV9_mutH3.1, the mutant histone H3.1-containing recombinant adenoassociated virus serotype 9. (**C**) Schematic representation of the final insert synthesized and cloned into a recombinant AAV9. This cassette encoded mutH3.1 flanked by loxP sites (purple), three single-guide RNAs (sgRNAs; blue) driven by the human polymerase III U6 promoters (green), and a mCherry fluorescent protein (red). In this approach, S10A point mutation would be introduced into only cre-expressing neurons (i.e., into Pdyn-expressing neurons in Pdyn::cas9-EGFP hybrids). CMV, human cytomegalovirus (CMV) immediate early enhancer and promoter; CBH, chicken beta-actin promoter with CMV enhancer. For sgRNA sequences targeting wild-type histone H3.1, see Appendix A. See also Appendix A for further technical details. (**D**) 3D volume reconstruction of micro-CT images, used for validating the intrathecal position of the inserted cannula before the osmotic pump implantation. The intrathecal catheter (arrow) is shown within the subarachnoid space in a living deeply anesthetized Pdyn::cas9-EGFP mouse. The catheter was introduced at the level of L5-L6 vertebral laminae and pushed up to L1-L2. Scale bar, 5000 μm. (**E**) In contrast to the hippocampus (hc), mCherry-specific RT-PCR produced a single sharp band in the spinal cord (sc) sample of a wild-type mouse that had been transfected with the AAV9_mutH3.1. GAPDH was amplified in both samples. BenchTop 100 bp DNA ladder was used as a reference.

**Figure 3 ijms-23-03178-f003:**
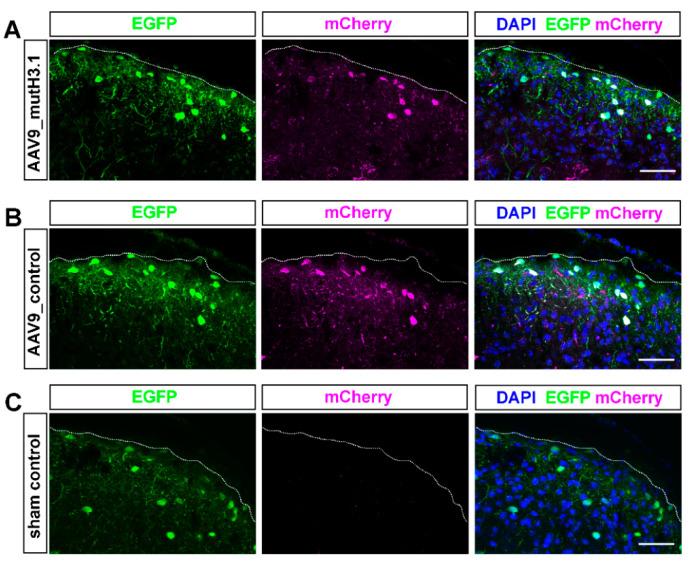
Distribution of AAV9 viral particle in the spinal dorsal horn of Pdyn::cas9-EGFP mice as identified by the presence of mCherry tag encoded by the virus. Representative images showing immunostaining with antibodies against EGFP (green), mCherry (magenta), and DAPI (blue) in projected images of 10 optical sections (each 0.5-µm-thick) taken with a 40× lens in the spinal cord from an AAV9_mutH3.1 vector treated (**A**), an AAV9_control treated (**B**) and a sham-operated (**C**) Pdyn::cas9-EGFP mouse. (**A**) Neurons showing mCherry-immunoreactivity are scattered throughout the SDH. Some Pdyn neurons (green to to their EGFP expression) show strong mCherry signal especially in the superficial region of the dorsal horn. (**B**) Administration of AAV9_control virus into Pdyn::cas9-EGFP mice produced an expression pattern of mCherry, similar to that shown in panel A. (**C**) The mCherry-specific fluorescent signal is completely missing in the transverse spinal cord sections of animals in the sham-operated group. White dotted lines indicate borders between white and gray matter. Scale bars is 50 µm.

**Figure 4 ijms-23-03178-f004:**
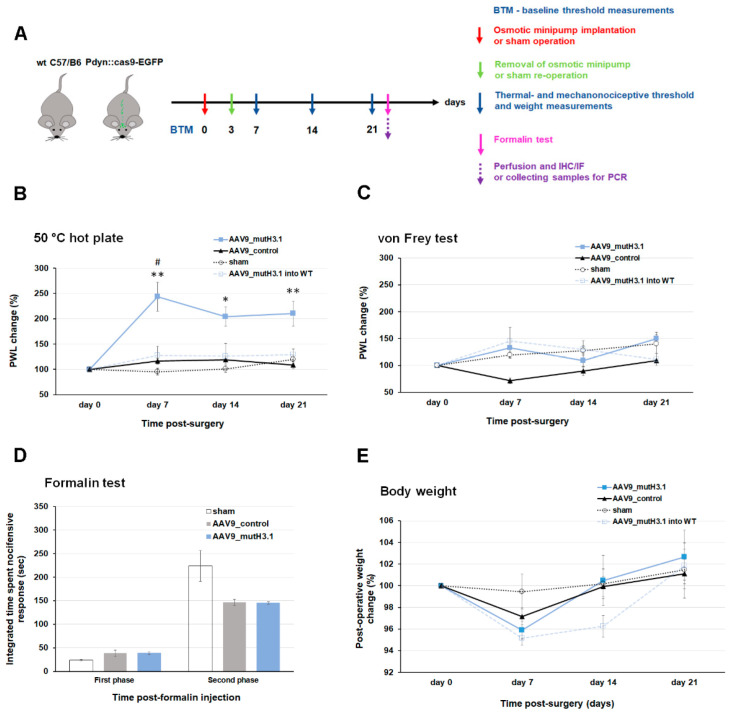
Intrathecal administration of AAV9_mutH3.1 into Pdyn::cas9-EGFP mice increases the thermal nociceptive threshold. (**A**) Schematic time scale of the experimental procedures. WT, wild-type C57Bl/6; IHC, immunohistochemistry; IF, immunofluorescent staining. (**B**,**C**) Changes in paw withdrawal latencies (PWL) to thermal- and mechanical pain were evaluated before (day 0) and after the surgery (day 7, 14, 21) in different groups of Pdyn::cas9-EGFP mice (i.e., AAV9_mutH3.1, AAV9_control, sham-operated) and in wild-type animals transduced with the AAV9_mutH3.1. Values at Day 0 represent the pre-surgery baseline values (BTM). (**B**) AAV9_mutH3.1-treated animals exhibited higher thermal nociceptive threshold compared to the AAV9_control and sham operated groups at day 7. This significant elevation was persistent until the end of the observational period. * *p* < 0.05 and ** *p* < 0.01 compared with the AAV9_control group (details in Appendix A). ^#^
*p* < 0.01 when the overall influence of the treatment with the AAV9_mutH3.1 on paw withdrawal latency (PWL) in response to noxious heat compared with the pre-surgery baseline (*p* = 0.009, *n* = 7, Kruskal–Wallis ANOVA). (**C**) Paw withdrawal latency to painful mechanical stimuli showed no significant alterations within, and differences between the groups. (**D**) Formalin-induced somatic pain was quantified as the integrated time spent exhibiting nocifensive behavioral during early (0–15 min) and late (15–60 min) phases of formalin application. Formalin-induced nocifensive behavior was reduced by 40% in the second phase in mice that had been infected with AAV9 irrespectively from their transgenes to be expressed, although, this reduction did not reach a statistically significant level. For additional details for statistical comparison see Appendix A. See also Appendix A for raw data of the formalin-induced nocifensive behavior. (**E**) Changes in body weight were evaluated before (day 0) and after the surgery (day 7, 14, 21) in different groups of Pdyn::cas9-EGFP mice (i.e., AAV9_mutH3.1, AAV9_control, sham-operated). Day 0 represents the pre-surgery baseline value. Transduction with the viruses (AAV9_mutH3.1 or AAV9_control) led to a modest body weight loss by day 7 that resolved later in all groups.

## Data Availability

Not applicable.

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
