# Peer review of "CRISPR/Cas9-Based Mutagenesis of Histone H3.1 in Spinal Dynorphinergic Neurons Attenuates Thermal Sensitivity in Mice"

_ijms, 2022, doi:10.3390/ijms23063178_

Round 1

Reviewer 1 Report

In this study, Mészár et al. assessed the contribution that histone H3.1 phosphorylation at position 10 serine in dynorphin+ neurons makes to thermal sensitivity. To this end, they developed a novel approach to mutate the serine at position 10 of H3.1 into alanine using a Cre-dependent CRISPR/Cas9 strategy delivered to the spinal cord via the intrathecal AAV administration. They performed a series of behavioral tests, with the relative sham and viral controls, to address the sensitivity to thermal and mechanical stimulation following the disruption of H3.1 S10 phosphorylation-dependent pathway. The authors concluded that phosphorylation at serine 10 of the histone H3.1 in dynorphin neurons is necessary for heat thermal sensitivity but not for mechanical sensitivity. Although this approach for inducing neuron-specific changes of epigenetic modifications is very interesting, and this work better clarifies the contribution of dynorphin neurons to sensory perception, and the controls for in vivo behavioral tests are well-designed, this study currently needs some additional analyses before publication, to strengthen some of the findings and the approaches.

  • It would help the interpretation of the behavioral results to insert two key quantifications in this manuscript:
    • The number of Pdyn::cas9-EGFP neurons that express Pdyn compared to the total number of Pdyn::cas9-EGFP neurons, and the number of Pdyn neurons that express Pdyn::cas9-EGFP compared to the total number of Pdyn neurons. This information would be important to define what is the actual percentage of dynorphinergic neurons that is being epigenetically modified.
    • Of similar importance would be to quantify the percentage of Pdyn::cas9-EGFP that express mCherry (and thus are effectively targeted by the CRISPR/cas9 strategy) following the AAV transduction.
  • The last paragraph describing in the details the efficiency of dorsal interneuron transduction (2.7) would be better placed after Paragraph 2.2. In this way, the reader would be shown all the expression data before diving into the behavioral characterization. Furthermore, to strengthen the claims of the paper and demonstrate that histone H3.1 serine 10 phosphorylation is indeed impaired by the Cre-dependent CRISPR/Cas9 strategy, authors should show the lack of phosphorylation of H3.1. at the S10 position in dynorphin neurons in the mice transduced with AAV9_mutH3.1 compared to AA9_control.
  • For the behavioral tests, in my opinion it would help the better understanding of the impairments to show the absolute values of latency to respond to both thermal and mechanical stimuli, in addition to just showing the percentage of changes as quantified in Figure 3. This information could go in the supplemental files.

Reviewer 2 Report

Dorsal horns receive sensory information from primary afferents and transmit it to several brain regions. Dynorphinergic neurons are considered major inhibitory neurons participating in pain processing. Mészár and colleagues hypothesized that phosphorylated serine 10 (S10) of histone H3.1 played an important role in central sensitization and heat hypersensitivity. The authors combined CRISPR/Cas9 and dominant-negative methods to perturb mouse H3.1 function in the dynorphinergic neurons of spinal dorsal horns. The mice showed increased thermal nociceptive threshold without affecting mechanosensation or acute chemonociception. The study can be improved with the following specific points being appropriately addressed.

  1. Multiple gRNAs targeting single genes often generate deletion between two cutting sites. Did you confirm the mutagenesis with sequencing? What percentage of indels and deletions was in mutant mice?  
  2. Figure 1 showed that Pdyn neurons were also located at the cortex, striatum, and inferior colliculus. Was AAV transduced into these regions? Can you show AAV infection in the brain regions? The somatosensory cortex encodes incoming sensory information from receptors. Did you observe Pydn neurons in the somatosensory cortex? 
  3. Did you observe the neuronal loss of Pdyn neurons? Did you check the early time point before viral transduction and compare them with the time point after AAV infection to confirm neuron survival? 
  4. Please also include the individual fluorescent channels for figure 4 to better visualize the viral transduction and overlap of Pydn neurons and AAVs with mutant histone H3.1.
  5. Did you confirm the S10H3 downregulation with supporting experiments, such as Western blot, qPCR, or immunostaining? Did you also check the changes of downstream targets of H3.1?
  6. Please provide more information about AAVs that were delivered. How much total AAVs did you transduce? 

Round 2

Reviewer 1 Report

The authors did a great job in addressing my criticisms and performing the additional experiments and analyses, as needed.

My only suggestion is to add as Supplementary Figure the data showing the reduced burn injury-induced S10H3 phosphorylation, as it is a nice experiment to prove the validity of their approach.
